# Decoupling Transmission and Transduction for Improved Durability of Highly Stretchable, Soft Strain Sensing: Applications in Human Health Monitoring

**DOI:** 10.3390/s23041955

**Published:** 2023-02-09

**Authors:** Ali Kight, Ileana Pirozzi, Xinyi Liang, Doff B. McElhinney, Amy Kyungwon Han, Seraina A. Dual, Mark Cutkosky

**Affiliations:** 1Department of Bioengineering, Stanford University, Stanford, CA 94305, USA; 2Department of Mechanical Engineering, Stanford University, Stanford, CA 94305, USA; 3Department of Cardiology, Lucile Packard Children’s Hospital, Stanford University, Stanford, CA 94305, USA; 4Department of Mechanical Engineering, Seoul National University, Seoul 08826, Republic of Korea; 5Department of Biomedical Engineering, KTH Royal Institute of Technology, 11428 Stockholm, Sweden

**Keywords:** large deformation strain sensor, implantable sensor, microelectromechanical system (MEMS), cardiac sensor, stretchable sensor

## Abstract

This work presents a modular approach to the development of strain sensors for large deformations. The proposed method separates the extension and signal transduction mechanisms using a soft, elastomeric transmission and a high-sensitivity microelectromechanical system (MEMS) transducer. By separating the transmission and transduction, they can be optimized independently for application-specific mechanical and electrical performance. This work investigates the potential of this approach for human health monitoring as an implantable cardiac strain sensor for measuring global longitudinal strain (GLS). The durability of the sensor was evaluated by conducting cyclic loading tests over one million cycles, and the results showed negligible drift. To account for hysteresis and frequency-dependent effects, a lumped-parameter model was developed to represent the viscoelastic behavior of the sensor. Multiple model orders were considered and compared using validation and test data sets that mimic physiologically relevant dynamics. Results support the choice of a second-order model, which reduces error by 73% compared to a linear calibration. In addition, we evaluated the suitability of this sensor for the proposed application by demonstrating its ability to operate on compliant, curved surfaces. The effects of friction and boundary conditions are also empirically assessed and discussed.

## 1. Introduction

The ability to accurately sense large deformations of soft structures is becoming increasingly valuable in various fields, such as soft robotics, wearables, textiles, and implantable medical devices. Despite this interest, there have been few demonstrations of highly stretchable sensors that provide consistent and repeatable results over many cycles. Conventional strain sensors, such as semiconductor and piezoresistive gauges and microelectromechanical systems (MEMS), can be manufactured with high precision for robustness and are often used in industrial applications that require long cycle lives. Nevertheless, they are limited to measuring small strains and are typically manufactured out of rigid components, inhibiting their use in applications with large deformations and soft structures. In order to overcome these limitations, recent research has focused on the development of highly-stretchable, large-deformation strain sensors using novel materials, including but not limited to ionic hydrogels, conductive polymer composites, and liquid-metal-in-rubber [1,2,3,4]. These methods can produce sensors with impressive signal-to-noise (SNR) ratios, but they often suffer from baseline conductive drift and unreliable interconnects between soft and rigid conductive components that limit their cyclic durability, having stable performance for up to 60,000 cycles at best [5,6,7,8,9]. Longer cycle life has been demonstrated but with evidence of sensor drift and changes in signal amplitude over time [10,11].

Other groups have developed large-deformation strain sensors by patterning inextensible conductors with rationally designed geometrical structures that impart stretchability (e.g., with a serpentine pattern), but this approach only permits stretch in a particular direction [12,13]. Moreover, there is a large mismatch in bulk material stiffness between the conducting material and the soft substrate that can lead to undesirable local stress concentrations, particularly when stretch is perpendicular to the preferred direction of electrode patterning. This can inhibit the sensing of soft structures that stretch multidimensionally, which is often the case for human health and biomechanical monitoring applications. It is also worth noting that most of these sensors are intrinsically sensitive to pressure and bending, leading to difficulties in discerning between mechanical phenomena [9].

Finally, others have attempted stretchable strain sensors that leverage the patterning or structural forming (e.g., wrinkling) of soft conductive materials, bypassing the limitation of mismatched mechanical properties but still leveraging the advantage of geometry-imparted stretchability for signal generation [9,14]. However, the actual integration of such sensors remains challenging, and the development of robust, miniaturized signal amplification, readout circuitry and shielding mechanisms for practical applications of these sensors has yet to be demonstrated.

In these traditional strain-sensing approaches, the transduction mechanism relies on the extension of the sensing unit, which results in a fundamental trade-off between mechanical properties and electrical integrity under repeated large deformations. To overcome this trade-off, we propose a large-deformation strain-sensing paradigm that decouples extension and signal transduction into separate components using a mechanical transmission system. Specifically, this approach leverages a non-conductive, soft elastomeric transmission that relays a mechanical signal to a high-sensitivity, robust MEMS transducer. The elastomeric transmission element converts high-strain, low-stress mechanical energy into low-strain, moderate-stress energy through the MEMS transducer, which converts the mechanical energy into electrical energy for communication. Since the transmission element does not have to conduct electricity, it can be optimized solely with respect to its mechanical properties. This allows for the electrical components to be compact and enables the use of mass-produced, highly-engineered electrical transducers with on-board amplification and communication systems. For some of the same reasons, MEMs transducers have been used in tactile and force sensing applications, such as robotic surgical grippers [15,16,17]. Overall, this approach offers improved performance and integration capabilities for practical applications that involve soft materials and large strains.

In the following sections, we describe the sensing principle in detail and demonstrate how this decoupling of mechanisms provides several advantages, such as design tunability, selective sensitivity, and high SNR. It also leads to comparatively high durability, making the sensor suitable for demanding applications that require millions of cycles. Given the sensor’s high mechanical lifetime under large deformation in particular, we investigated its application as an implantable cardiac strain sensor, where it must withstand millions of heartbeats at high strains but provide robust and continuous communication.

## 2. Sensor Concept and Application

### 2.1. Sensing Principle

The proposed sensor uses a transmission and transducer pair to measure large strains (Figure 1). The transmission element couples the mechanical signal of interest to a robust, high-sensitivity transducer that converts the mechanical signal into an electrical signal (Figure 1A–C). The transmission element can be visualized as a spring with some stiffness, and the transduction element can be visualized as a low-deformation normal force sensor that is mechanically attached to one end of the spring (Figure 1A, right). As the spring is stretched, the normal force at the proximal end increases in proportion to the spring’s stiffness, and that force is transduced into an electrical signal. By decoupling the transmission and transducer components, each can be optimized independently for its specific role in the system. The transmission can be designed to maximize its stretchability and mechanical consistency, without being concomitantly constrained with conductivity requirements. Meanwhile, the transducer can be optimized for efficient mechanical to electrical energy conversion, robust packaging, and integrated communication protocols, without simultaneous deformation requirements.

### 2.2. Application

As discussed in the Introduction section, the proposed sensor can find a range of applications spanning soft robotics to implantable devices. We believe the latter field has encountered the greatest challenges with respect to durability, complicated by the requirement for material biocompatibility, given the intimate interaction with human tissues. Thus, we have identified human health as the segment with the highest need for technical solutions. In response to this clinical challenge, we demonstrate the design and assessment of this strain sensing paradigm in the context of a cardiovascular application, namely, implantable cardiac functional monitoring. Direct cardiac mechanical monitoring has challenged the durability of conventional mechanical sensors due to the number of cycles of the beating heart (40 M cycles/year) at high strains (15–20%). Additionally, this particular application bears requirements for mechanical and material biocompatibility, given the soft nature of live heart muscle. Nevertheless, direct mechanical metrics of heart function are known to be clinically useful in predicting and monitoring the state of cardiac health, particularly in patients with known cardiac disease. Clinical research has established that imaging-derived measures of global longitudinal strain (GLS) of both right and left ventricles are informative and predictive measures of cardiac health [18,19,20]. Such functional monitoring can be particularly valuable in post-surgical settings, such as a post-mitral valve replacement, left ventricular assist device (LVAD) implantation, or heart transplantation [21,22,23]. Nevertheless, echocardiographic imaging methods traditionally used to obtain GLS measurements require an expert clinician, and measurements are person-dependent, inhibiting their use in remote settings; moreover, these measurements are intermittent rather than continuous and may miss critical changes in cardiac function [11]. The development of a robust and accurate compliant strain sensor that can be implanted during the time of surgery and continuously monitor cardiac GLS after the patient leaves the hospital would enable physicians to monitor patients’ health remotely, optimize treatment, and predict adverse outcomes in real-time. A visualization of the proposed embodiment (on the right ventricle, for example) is shown in Figure 1D. Towards the development of a sensor that can be practically implemented, we demonstrate the proposed sensor’s ability to perform under the unique and demanding constraints of such an application, as listed here:Soft: comparable to the stiffness of native heart tissue (50 kPa [24]).Durable: consistent signal over at least a million cycles (corresponding to about 11 days at 60 beats per minute).Selectively Sensitive: insensitive to non-uniaxial strain signals that might occur, such as bending from the heart’s torsion or pressure from external organs.Conformable: consistent signal on a dynamic and compliant curved surface.Extensibile: achievable strains of at least 20% with a corresponding stable and predictable signal [25].Dynamic: consistent and accurate signal at various heart rates and under dynamic load profiles that contain various frequency components.Biocompatible: made of non-toxic materials that can be implanted with minimal inflammatory response from the tissue.

## 3. Sensor Design and Fabrication

### 3.1. Application-Specific Design Selections

The working principle of this sensor allows for a broad choice of transmission and transduction elements. Given the requirements of a sensor that aims to be used for implantable applications, elements were selected that are biocompatible, robust, and miniaturizable. For the transduction element, we chose a high-precision MEMS barometric pressure sensor (BMP384, Bosch Sensortec), consisting of a transduction mechanism that relies on the deflection of a capacitive diaphragm. This selection was inspired by Tenzer et al., who developed a tactile sensor that used elastomeric potting to transform a MEMS barometer into a compliant force sensor [16]. MEMs barometers are commercially available and industrially manufactured, ensuring high robustness and low noise. Further, these sensors come with integrated, on-board amplification and communication protocols, such as I^2^C, enabling a simple and compact readout system. For the transmission element, we chose an elastomer with demonstrated biocompatibility, namely, silicone (Ecoflex, Smooth-On), in the Shore 00 range, which corresponds to a range of stiffnesses comparable to that of human tissue [26,27,28]. Moreover, this material is isotropic and homogeneous, leading to directionally independent mechanical properties and minimal hysteresis compared to composite materials [29]. Finally, we designed the geometry of the elastomeric transmission element to be as small as possible but with a large enough cross-sectional area (w and l) and length (l) to completely cover the area of the pressure sensor (3 mm × 3 mm) and span the height of a ventricle (5 cm), respectively.

### 3.2. Sensor Fabrication

The sensor can be fabricated in four steps, as illustrated in Figure 2. First, a printed circuit board (PCB) was designed and manufactured (OSH Park, Lake Oswego, OR, USA), mounting the pressure sensor and ancillary electrical components. A custom 3D printed mold was designed and manufactured to hold the PCB in place and provide a form to cast the silicone transmission element so as to completely encapsulate the pressure sensor. To optimize the sensitivity of the sensor, the entire assembly was degassed, removing the air inside the MEMS chip and allowing the silicone prepolymer to completely infiltrate the protective casing and bond directly to the capacitive membrane of the pressure sensor [16]. The assembly was left to cure, and once the sensor was removed, wires were soldered for I^2^C communication to an off-board ATmega 328 microcontroller (Arduino, New York, NY, USA). The entire process, excluding the time required to manufacture the PCB and print the mold, takes about thirty minutes and is largely dependent on the cure time of the polymer. Additionally, the entire assembly costs less than fifty US dollars.

## 4. Sensor Characterization

### 4.1. Quasi-Static Characterization

To confirm the sensing principle, the sensor was characterized in a quasi-static condition at a consistent and controlled frequency of 1 Hz, corresponding to a typical resting heart rate of 60 beats per minute (bpm). The sensor was stretched uniaxially using a dual-mode muscle lever with programmed displacement control (Aurora Scientific 309C). Multiple sensors were fabricated out of elastomers of various hardnesses for comparative analysis. As the transduction element is a pressure transducer, the signal is reported in pressure units (kPa). Further, because the transmission element pulls the diaphragm away (see Figure 1B), a tensile strain results in a pressure decrease from baseline (atmospheric pressure); however, Figure 3A,C reports absolute changes in pressure. Uniaxial testing results in Figure 3A demonstrate that strain and pressure are linearly correlated (R^2^ = 1 for all hardnesses) and that the sensitivity of the signal can be increased by increasing the hardness, or correspondingly the Young’s modulus, of the transmission element. Overall, there is a general trade-off between stiffness and sensitivity. This makes intuitive sense, as the stress on the capacitive diaphragm will increase in proportion to the stiffness of the elastomeric element.

The sensor displays an impressive signal-to-noise ratio (SNR), even with the softest elastomer. Specifically, the sensor composed of the 00–10 shore hardness elastomer has a sensitivity of 136 Pa per percent strain, and the transducer has an unfiltered RMS noise of 1.2 Pa, yielding an SNR of over 100. As the Shore hardness of 00–10 (empirically determined to be 7 kPa) is softer than cardiac tissue yet still provides sufficient signal, we chose to continue analysis with this.

Figure 3B illustrates the sensor’s response under cyclic loading. Cycles 1, 2, 3, and 3600 were comparatively evaluated to assess signal consistency, and the raw data are plotted, illustrating the spread of the sensor readings. The average standard deviation between cycles was 0.013 kPa. The sensor displays moderate but consistent hysteresis. This result was expected, given that silicone is a viscoelastic material. Figure 3C shows the sensor’s signal under an arbitrary range of sinusoidal strain amplitudes.

### 4.2. Decoupling Undesirable Signals

In uncontrolled settings, it is likely that a sensor will experience environmental changes and mechanical loads that could influence the signal and confound sensor readings. This is particularly true in the proposed application of a cardiac GLS sensor. Even though the sensor should only provide a signal for unidimensional strain in the longitudinal direction, the heart creates a challenging environment. Upon implantation, the sensor will be submerged in a primarily aqueous environment, and, although the range of body temperature is fairly limited (36–37 ${}^{{}^{\circ}}$C), previous analysis of a silicone-potted MEMS sensor has demonstrated significant temperature sensitivity [16]. The sensor’s responses to these environmental factors were evaluated, and results are depicted in Figure 4A,B.

Figure 4A illustrates the raw response of the static sensor assembly being heated across a range of physiological temperatures. The relationship between pressure and temperature appears linear (R${}^{2}$ = 1). The result matches well with previous work that also observed a linear relationship between sensor readings and pressure when temperature was increased, and in this work it was hypothesized that the bonding of silicone to the internal silicon diaphragm of the MEMS likely influences the composite thermal coefficient of expansion and could partially account for the observed effect [16]. Sensor drift due to temperature can be easily compensated for with the onboard temperature sensor of the BMP384 [30]. To evaluate the sensor in an aqueous environment, the silicone transmission was left submerged in water over 18 h and subsequently tested. Figure 4B plots the sensor response for both the dry and submerged sensors, indicating that water has little effect on the mechanical properties of the silicone.

In the intended application, the sensor could experience contact pressure from nearby organs and experience non-longitudinal loading, primarily associated with bending, as the heart twists during contraction. The sensor’s responses to these scenarios were evaluated. The sensor was subjected to bending up to 45 degrees, as illustrated in Figure 4C. The sensor was bonded to a glass plate in two places, at the PCB end and the distal end of the transmission, and the distal end was displaced along an arc of radius equal to the sensor’s length. The bending test was conducted four times, and Figure 4C (top) displays a representative trace. The sensor was held at 45 degrees for approximately 3 s; averages and standard deviations are illustrated in the bottom graph of Figure 4C. On average, bending resulted in a 0.215 kPa signal response with a standard deviation of 0.046 kPa. This displacement-driven loading condition is dominated by bending but may include small amounts of stretch, compression, and/or shear. Hence, we expect a small change in the transducer signal, as confirmed in the plot.

To evaluate the sensor’s response to external pressure, a 100 gram weight was placed on the silicone transmission using a glass slide to distribute the load. This pressure test was conducted four times, and Figure 4D (top) displays a representative trace. The weight was placed on the sensor for approximately 3 s for each trial; averages and standard deviations are illustrated in Figure 4D bottom graph. On average, the added weight resulted in a −0.230 kPa signal response with a standard deviation of 0.045 kPa. The loading condition may produce a small amount of stretch near the proximal end due to the deformation caused by the edge of the glass slide, evidenced by a slight change in transducer signal in Figure 4D.

### 4.3. Durability

We investigated the durability of the proposed sensor under long-term cyclic strain. Specifically, the sensor was loaded with a sinusoidal displacement of 30% strain at 20 Hz, corresponding to 1.5 cm of extension. Figure 5 illustrates the sensor’s response over 1,000,000 cycles. There is a slight baseline drift from cycle 100 to cycle 1,000,000 of about 109 Pa. This could be attributed to a change in temperature that is smaller than the manufacturer-defined absolute accuracy of the onboard MEMS pressure sensor (0.65 °C) [30]. Given our empirically defined calibration (Figure 4) of 249 Pa per degree Celsius, a change of 0.44 °C could account for such a change in pressure.

Signal drift that occurs in conventional stretchable sensors during cyclic mechanical loading can be attributed to many factors, a main one being the rearrangement of conductive particles in the soft, polymeric matrix, which can result in baseline resistive drift and even complete loss of conductivity [31]. Selection of a pure, non-conductive polymer as the deforming element eliminates the risk of signal drift due to particle migration. Moreover, elastomers of a single polymer network exhibit less hysteresis and are generally more elastic than filled polymers, which may reduce mechanical creep compared to composite elastomers [29]. Finally, the integrated electronics of the MEMS pressure chip reduce the risk of mechanical and electrical interconnect instability, improving the cycle-life of the sensor. These features likely contribute to the enhanced signal stability and drift-free nature of the sensor.

### 4.4. Dynamic Characterization

Although a quasi-static characterization is useful for demonstrating the basic function of the proposed sensing principle, practical applications of medical and in-body strain sensors involve varying frequencies. The sensor was sinusoidally strained across a range of physiologically relevant frequencies, ranging from 0.5 to 2 Hz. Figure 6A plots sensor signal as a function of strain, demonstrating a slight frequency dependence of the sensor that cannot be captured by a quasi-static characterization. Not only is there a hysteretic component to the response, but the sensitivity, or slope, of the signal increases by as much as 25% at the highest swept frequency. This is illustrated by the pressure-strain response rotating counter-clockwise about the origin. As the pressure signal is directly linked to the mechanical behavior of the transmission element, this response can be attributed to the time-dependent viscoelastic nature of the silicone. Without a proper compensation or mapping algorithm, it would be impossible to differentiate between a change in frequency at a constant amplitude and an increase in strain amplitude at the same frequency. This is important for cardiac applications, as heart rate can vary over the course of minutes; moreover, a single heart beat can contain many strain-rate components, as total systolic contraction occurs over a fraction of the whole heartbeat.

The use of model-based transfer functions has been demonstrated to effectively compensate for hysteresis and strain-rate dependent material behavior in elastomeric sensors [32]. To enable dynamic mapping of sensor pressure signal to ground-truth strain, a transfer function based on a multi-element rheological model that represents dynamic stiffness of an elastomeric transmission element was constructed [33]. Rheological models are typically composed of both spring and damper elements to represent the elastic and viscous contributions of a viscoelastic material, respectively. Specifically, we chose a standard generalized Maxwell model (GMM) to represent our system, depicted in Figure 6B. Following [34,35], the dynamic stiffness, or stress over strain, of a GMM can be modeled in the frequency domain as follows:(1)$$Z\left(\omega \right)=K_{0}+\sum_{i=1}^{N}\frac{j\omega K_{i}C_{i}}{K_{i}+j\omega K_{i}C_{i}}$$
where $K_{0}$ is the static stiffness and $K_{i}$ and $C_{i}$ are the stiffness and damping values for the Maxwell element *i*. This can be converted into the equivalent pole-zero formulation:(2)$$Z\left(\omega \right)=K_{0}\prod_{i=1}^{N}\frac{1+(j\omega /\omega_{z,i})}{1+(j\omega /\omega_{p,i})}$$

However, pressure will be considered the input to the system and mapped to strain, so the transfer function can be thought of as a model for dynamic compliance as opposed to stiffness. Therefore, the transfer function G(ω) is defined as
(3)$$G\left(\omega \right)=\frac{1}{Z(\omega )}=\frac{1}{K_{0}}\prod_{i=1}^{N}\frac{1+(j\omega /\omega_{p,i})}{1+(j\omega /\omega_{z,i})}$$

Thus, the transfer function variables that can be tuned are the number of zero-pole pairs (*N*), each of which represents a single Maxwell element, and the values of $w_{z,i}$ and $w_{p,i}$ in (Equation 2), which are related to the stiffness and damping of the system. Using readily available MATLAB functions such as tfest(), we fit a transfer function to map the pressure output to the ground-truth frequency sweep for a varying number of Maxwell elements (*N* = 1, 2, and 3). The raw data were modified slightly to facilitate the model fitting process. Specifically, the raw pressure waveform from the frequency sweep was converted from Pa to kPa and multiplied by −1, given that pressure decreases as strain increases, and both signals were detrended. A visualization of the modified empirical data is presented in Figure 6C. A zero-order mapping refers to the linear calibration made in the quasi-static characterization (Figure 3A).

The MATLAB function tf2zp() was used to determine the values for the gain, poles, and zeros from the transfer function obtained from tfest(), which outputs a ratio of polynomials. The gain is equal to the inverse of stiffness $K_{0}$, following from (Equation 3), with units of (% Strain)/kPa. Table 1 presents the identified values for each Maxwell model order and their respective errors on the validation data. The errors suggest that two Maxwell elements are sufficient to comprehensively capture the viscoelastic transmission response. The second-order model’s response is plotted in Figure 6D overlaid on the input data. Overall, the second-order TF mapping demonstrates excellent hysteresis and frequency-dependent stiffness compensation to yield predicted strain values that map to ground-truth with an average error of 0.3% strain, or 1.76% error. Although the gain, poles, and zeros are sufficient to uniquely describe the system, a mathematical relationship between the zeros and poles and the constitutive parameters $K_{i}$ and $C_{i}$ in Equation (Equation 1) is described in Renaud et al. [34] for the interested reader.

## 5. Application-Specific Testing

### 5.1. Strain Sensing of a Realistic Heartbeat

To validate the models on physiologically realistic test data, the sensor was stretched at a strain-time profile that represented a true cardiac contraction. The strain-time profile was gathered from the literature and used to program muscle lever displacement [36]. The sensor was strained according to this displacement profile at 60, 90, and 120 beats per minute (bpm).

Figure 7A compares the errors in predicted strain for the zero, first, second, and third-order models for a range of heartbeats. The second-order model demonstrates a significant reduction in RMSE by 73%, averaged across all cases, whereas the first and second-order models reduced the error by 20% and 30%, respectively. Although results from the model fitting in Table 1 suggest that both second and third-order models will perform substantially better, the inability for the third-order model to correctly predict the test data is likely due to overfitting, providing compelling evidence for the choice of the second-order model. Figure 7B,C illustrate the response of the 90 bpm scenario for both second-order and zero-order (linear) mapping. The linear mapping shows clear hysteresis, and the second-order TF compensates effectively.

### 5.2. Dynamic Curvature

The sensor’s ability to conform to a heart-like surface and reliably measure strain over cyclic loading conditions was investigated in a benchtop setup. A ventricular phantom was modeled using a hollow prolate chamber design, which is a simple, widely used representation of ventricular geometry [37,38,39]. A negative mold of the prolate was 3D printed and subsequently cast with silicone of shore hardness 00–20, corresponding to a Young’s modulus comparable to that of the heart [27,28]. The prolate was mounted to a custom rig with attached tubing to allow for inflation and deflation of the chamber with air. The sensor was bonded with adhesive to the phantom at two points, the PCB end, and the tail end of the transmission element.

Figure 8A illustrates a critical mounting consideration for this application. To achieve efficient coupling between stretch and pressure on the diaphragm, it is necessary to ensure that the uniaxial stress is transmitted throughout the length of the stretchable element and not modified by counteracting forces or moments. In particular, frictional forces between the sensor and phantom surface can cause shear in the transmission element, attenuating the coupling between stretch and normal stress on the diaphragm in a complex manner (Figure 8B, Left column). Note that this phenomenon is amplified in our experimental test set-up, as the silicone-cast phantom has a higher coefficient of friction than the surface of the heart. To alleviate frictional forces, a commercially available, biocompatible hydrogel (Silvex, Los Angeles, CA, USA) was spread between the phantom surface and the sensor. The left and right columns in Figure 8B illustrate how friction can cause shear and affect pressure–strain coupling. Additionally, we cast the PCB into a small anchoring sleeve that was rigidly bonded to the phantom to prevent rotation.

The phantom was manually inflated and deflated with 70 mL of air, and the sensor signal was recorded for two cases, with and without hydrogel. The graph in Figure 8 depicts the results, clearly showing that the friction-reducing hydrogel increases the signal, likely due to enhanced mechanical coupling between overall stretch of the elastic element and strain at the transducer.

## 6. Discussion

This study illustrates the conception, fabrication, and testing of a novel, large- deformation strain-sensing mechanism. Highly stretchable strain sensors are typically affected by a trade-off between ideal mechanical and electrical properties: to achieve mechanical impedance matching to soft substrates, the cyclic durability of the conductive elements is compromised. We hypothesized that separating the transmission and transduction functions into separate elements would bypass this trade-off. We optimize the transmission using a highly-compliant, biocompatible elastomer; for the transducer we use a high-precision, robust MEMS transducer with integrated signal processing and digital communication. The soft transmission enables the coupling of large deformations to local variations in stress on the diaphragm of a commercial MEMS barometric pressure sensor. The pressure sensor is highly accurate and has provision for thermal compensation. Durability testing over one million cycles resulted in negligible signal drift and no apparent physical degradation, supporting the proposed hypothesis.

We further hypothesized that, by choosing a pure, homogeneous silicone elastomer, the response could be simply modeled with standard rheological frameworks. Using a lumped-parameter model, we evaluated a transfer function based on the generalized Maxwell model, comparing various model orders by assessing their strain-mapping accuracy on a test data set. The second-order transfer function performed the best, reducing RMSE by around 73% from the linear (zero order) model. The first-order model, on the other hand, only reduced RMSE by 20%, supporting the use of a second-order model, despite the added complexity. The third-order model appeared to overfit the data, highlighted by its comparatively poor performance on the test set, only reducing error by approximately 30%. Sensor response can be generally tuned by selecting $K_{0}$, a function of the Young’s Modulus and cross-sectional area of the transmission; this was demonstrated by fabricating and testing sensors with transmission elements of various stiffnesses.

The proposed approach results in additional desirable attributes, including low-cost, readily available materials, low complexity of fabrication, and integrated digital communication. We encourage future work to consider other applications requiring sensors that achieve high strains with soft mechanical properties but need to prioritize high-durability and easy integration. Examples include soft-robotics, human–robot interactions, and human motion sensing.

A particular focus of this study was to demonstrate suitability of the sensor in a demanding, implantable application. We showcase a sensor specifically designed for implantation on the outside of the heart to continuously measure longitudinal strain. The sensor’s durability extends the usability of soft strain sensor technology beyond the immediate post-operative setting and towards long-term monitoring applications. Such a sensor could provide real-time information on heart function and inform timely treatment decisions, even when the patient has left the hospital setting. Towards this end, we demonstrated a proof of concept, where we evaluated the sensor’s ability to measure dynamic strain on a phantom with realistic geometry. These experiments highlight a critical consideration for implantable applications, namely, the mechanical transparency of the silicone element that ensures coupling to the MEMS transducer. Although the hydrogel was sufficient in the benchtop set-up to reduce shear-inducing frictional forces between the sensor and the phantom, it is important to address the differences in the mechanical environment that would exist in the in vivo scenario. Firstly, the hydrogel used in this study was a free-flowing gel which would likely be absorbed into tissue over time. Future work should investigate the use of durable, stretchable hydrogels based on polymeric networks, such as those used by Roche et al. in a relevant cardiac application [40]. Additionally, the heart will be surrounded by external organs that may externally press on the sensor and inhibit movement; however, the heart has a natural potential solution to this problem. To protect the heart and coronary vessels from constant rubbing against other organs, the heart is supported by the pericardium, an external lubricated sac. This sac would be surgically opened for sensor placement and subsequently closed with sutures. The authors speculate that the serum produced by the pericardium could facilitate the mechanical transparency of the sensor and should be a subject of future investigation. Though a promising proof-of-concept, the sensor’s true suitability for long-term implantability should be investigated in in vivo studies that evaluate physiological tissue responses, such as fibrosis and inflammation, which are other physiological factors that may impact sensor performance. Future iterations on this work will also investigate the integration of wireless communication protocols, which would substantially enhance the implantability of this technology.

## Figures and Tables

**Figure 1 sensors-23-01955-f001:**
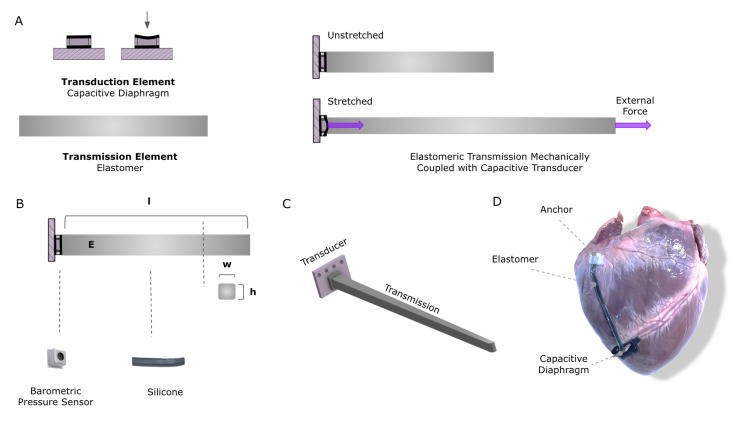
Sensing concept: (**A**) Depiction of transduction and transmission elements. The right panel illustrates how they are mechanically assembled to couple signal with uniaxial strain. (**B**) A MEMS barometer and soft silicone were chosen for the application. The transmission element is parameterized by the width, (**w**), and height, (**h**), of the cross-sectional area; and the length (**l**) and Young’s modulus (**E**) of the silicone. (**C**) Rendering of the complete sensor assembly. (**D**) Visualization of the sensor on the proposed application for measuring global longitudinal strain on the heart postoperatively.

**Figure 2 sensors-23-01955-f002:**
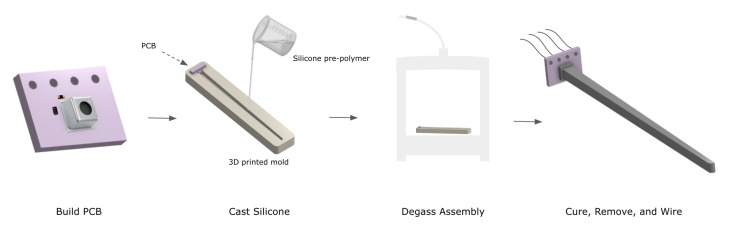
Sensor fabrication. The sensor is fabricated in a quick, facile manner. The PCB is potted with silicone of the chosen dimensions programmed into a negative 3D printed mold. The entire assembly is degassed and then left to cure. Wires are then soldered for I^2^C communication.

**Figure 3 sensors-23-01955-f003:**
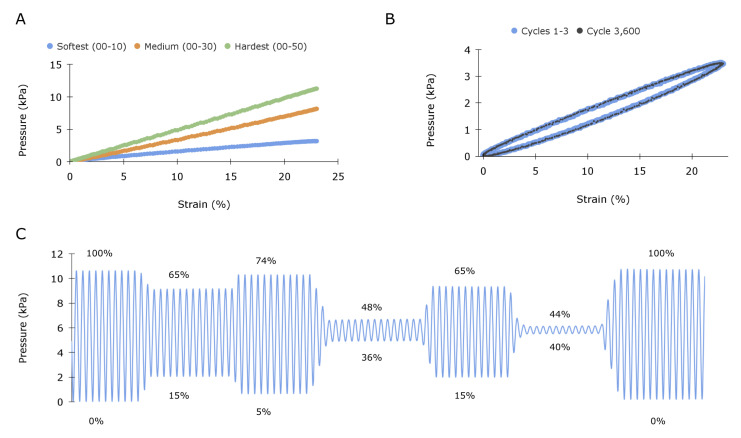
Static sensors’ characterization: (**A**) Characterization of sensors fabricated from various hardness elastomers, ranging from soft (Shore 00–10) to medium (Shore 00–30) to hard (Shore 00–50), demonstrating the tunability of sensor sensitivity. (**B**) Response of the 00–10 sensor subjected to a 1 Hz sinusoidal controlled displacement. Cycles 1–4 are plotted in blue, and cycle 3600 is plotted in black. The average standard deviation between each cycle is 0.013 kPa. (**C**) Demonstration of 00–10 sensor displaced to various amplitudes ranging up to 100% strain. Numbers above and below the plot lines quantify the peak and trough strain values for each section.

**Figure 4 sensors-23-01955-f004:**
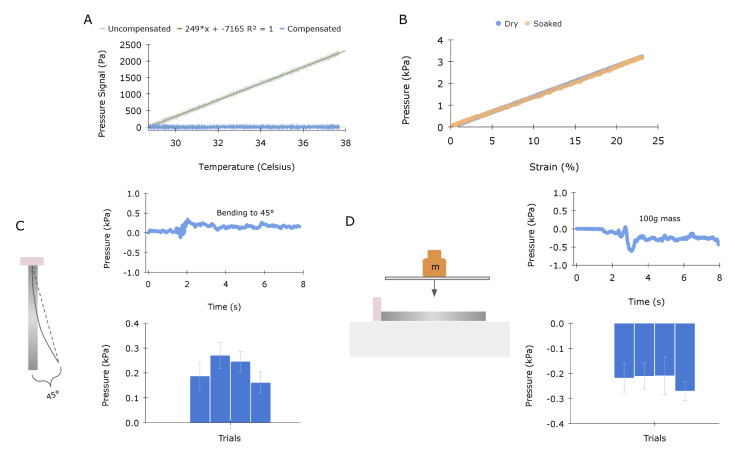
Signal decoupling and sensitivity characterization: (**A**) The uncompensated signal (gray) quantifies the sensor’s response to temperature changes, which appears linear (R^2^ = 1). Changes in temperature can be subtracted through a linear calibration using the onboard temperature sensor, as demonstrated by the compensated signal (blue). (**B**) The transmission element was allowed to soak in water for 18 h and then subjected to uniaxial cyclic loading. The graph demonstrates no change in output signal. (**C**) Sensitivity to bending, evaluated by rotating the tip by 45 degrees. Four trials were conducted, yielding averages of 0.215 and 0.046 kPa, respectively. The top graph illustrates a single trial over time, and the bottom graph displays each trial’s mean and standard deviation with error bars. (**D**) Sensitivity to normal pressure, evaluated by placing a 100 g weight on top of the sensor. Four trials were conducted, yielding an average and standard deviation of 0.230 and 0.045 kPa, respectively. The top graph illustrates a single trial over time, and the bottom graph displays each trial’s mean and standard deviation with error bars.

**Figure 5 sensors-23-01955-f005:**
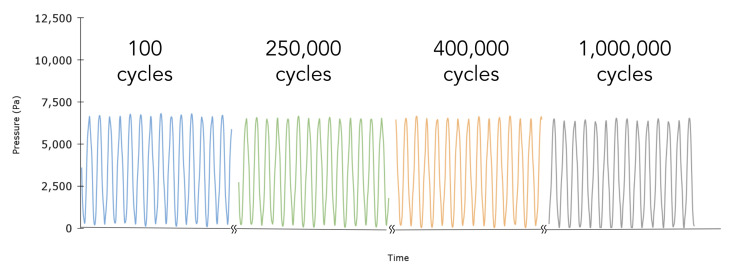
Durability analysis: Sensor durability was investigated through cyclic loading at 20 Hz for one million cycles. The output signal shows negligible drift.

**Figure 6 sensors-23-01955-f006:**
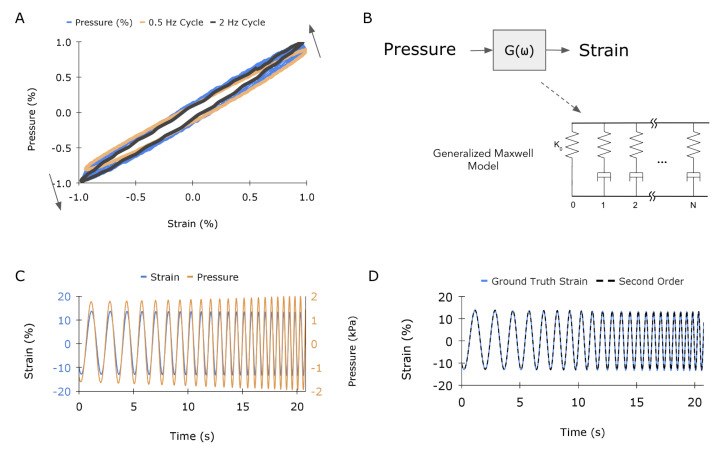
Sensor model for dynamic mapping: (**A**) Graph of the normalized pressure vs. normalized strain response across frequencies ranging from 0.5 to 2 Hz. The orange loop represents the slowest cycling rate, and the black loop represents the fastest. (**B**) Illustration of how a transfer function, G(ω), can be developed to map the input (pressure) to the output (strain). The model is parameterized by stiffness $K_{0}$ and the number of Maxwell elements (0 to N). (**C**) Illustration of the training data for the model, with pressure (blue) and strain (orange) as input and output, respectively. (**D**) Comparison of the strain for ground truth and second–order model.

**Figure 7 sensors-23-01955-f007:**
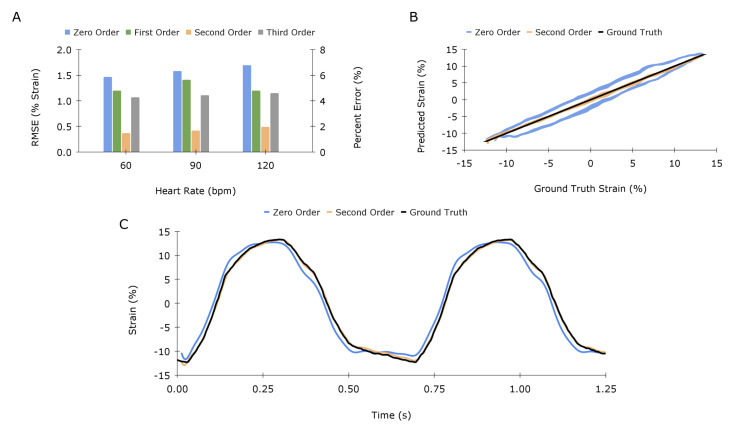
Application–specific model validation: (**A**) Quantification and comparison of the error in predicted strain for various heart rates for zero, first, second, and third-order models, displayed as percent strain and percent error of strain magnitude. (**B**,**C**) offer different visualizations of results for the 90-beats-per-minute case.

**Figure 8 sensors-23-01955-f008:**
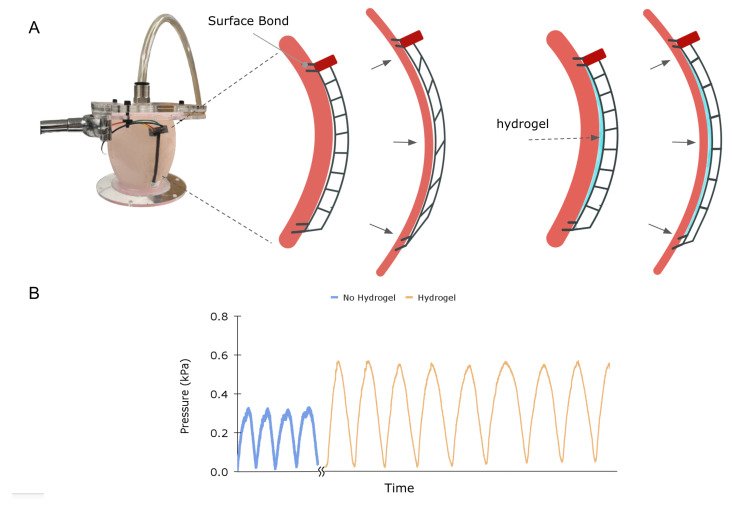
Strain sensor on a compliant, curved surface: (**A**) Left picture of the ventricular heart phantom benchtop setup. Middle: Illustration of phantom cross-section with a sensor demonstrating shear stress induced in the transmission element upon inflation. Right placement of hydrogel between the sensor and phantom reduces friction, eliminating shear and allowing for uniaxial stretch in the transmission. (**B**) Sensor signal for 70 mL inflation and deflation of the phantom with and without the hydrogel present.

**Table 1 sensors-23-01955-t001:** Transfer function parameters—gain ($1/K_{0}$), zeros, and poles, are reported for each model order fitting (N = 0, 1, 2, and 3), and percentage error of the model’s performance on the validation set is reported as a percent of the total strain signal amplitude.

Elements (N)	$1/K_{0}$	Zeros	Poles	% Error
0	7.35	n/a	n/a	5.48
1	6.32	−7.26	−5.32	2.25
2	6.17	−11.26, −1.75	−8.97, −1.43	1.76
3	7.14	−7 ×$10^{6}$, −9.33, −1.53	−6 ×$10^{6}$, −11.8, −1.85	1.80

## Data Availability

Data are available from the corresponding author upon reasonable request.

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
