# Peer review of "Decoupling Transmission and Transduction for Improved Durability of Highly Stretchable, Soft Strain Sensing: Applications in Human Health Monitoring"

_sensors, 2023, doi:10.3390/s23041955_

Round 1
Reviewer 1 Report
This paper reports a detailed research concerning the fabrication and assessment of high signal-to-noise (SNR) ratio, extreme durability, and tunable design strain sensors for human health monitoring by using a mechanical transmission system. The sensing principle and decoupling of mechanisms are clearly described and the results are discussed with competence. The obtained device could be really useful. The novelty of this work is the device, not the mechanism. However I believe this paper worthy for publication in the Sensors. Here, I just have only a recommendation of that the abstract and conclusion parts should include significant quantitative data if possible.
Reviewer 2 Report
This paper presents an interesting large-range strain sensor configuration by combining a stretchable element with a pressure transducer. The paper is well written, although it is heavy on the sales pitch and light on the science.
1. This is particularly true of the final discussion section, which should be re-written to enumerate scientific insights (with quantification to support each observation).
2. There is no statistical analysis of the test results – every result appears to show only a single test result, with none of the typical box plots or other methods of displaying the statistical spread of the results.
3. The assessment of temperature dependence should presumably include a simple analysis of whether the drift is caused entirely by thermal expansion.
4. The Maxwell model discussion is particularly lacking in details. It would be impossible to replicate the results. I would expect at least a table showing the spring / damper values for the 2-element model. I also found 6D counter intuitive, in the sense that I would naturally expect the predicted values to be on the x-axis (since there would be only a single value for each strain) and the true value on the y-axis (since there would presumably be 2 true values for each predicted value).
5. The final issue of how constrained the stretchable silicone element would be in an in-vivo setting could do with more discussion (although I appreciate that the authors have partially covered this with the discussion about friction and hydrogels).
6. Since the authors are specifically aiming to reduce drift, it would be valuable to compare creep rates of typical conductive viscoelastic sensors with the pure silicone stretchable element – at least in a brief discussion. I do believe that the pure silicone will creep much less, and the results appear to show that, but some discussion would be good.
Round 2
Reviewer 2 Report
The authors have addressed the reviewer's concerns very nicely.